# Adaptive Segmentation Algorithm for Subtle Defect Images on the Surface of Magnetic Ring Using 2D-Gabor Filter Bank

**DOI:** 10.3390/s24031031

**Published:** 2024-02-05

**Authors:** Yihui Li, Manling Ge, Shiying Zhang, Kaiwei Wang

**Affiliations:** 1State Key Laboratory of Reliability and Intelligence of Electrical Equipment, Hebei University of Technology, Tianjin 300130, China; 202131404090@stu.hebut.edu.cn (Y.L.); 202121401030@stu.hebut.edu.cn (S.Z.); 202131403029@stu.hebut.edu.cn (K.W.); 2Hebei Key Laboratory of Electromagnetic Field and Electrical Reliability, Hebei University of Technology, Tianjin 300130, China

**Keywords:** threshold segmentation, 2D-Gabor filter bank, subtle defect images, BP neural network classification

## Abstract

In order to realize the unsupervised segmentation of subtle defect images on the surface of small magnetic rings and improve the segmentation accuracy and computational efficiency, here, an adaptive threshold segmentation method is proposed based on the improved multi-scale and multi-directional 2D-Gabor filter bank. Firstly, the improved multi-scale and multi-directional 2D-Gabor filter bank was used to filter and reduce the noise on the defect image, suppress the noise pollution inside the target area and the background area, and enhance the difference between the magnetic ring defect and the background. Secondly, this study analyzed the grayscale statistical characteristics of the processed image; the segmentation threshold was constructed according to the gray statistical law of the image; and the adaptive segmentation of subtle defect images on the surface of small magnetic rings was realized. Finally, a classifier based on a BP neural network is designed to classify the scar images and crack images determined by different threshold segmentation methods. The classification accuracies of the iterative method, the OTSU method, the maximum entropy method, and the adaptive threshold segmentation method are, respectively, 85%, 87.5%, 95%, and 97.5%. The adaptive threshold segmentation method proposed in this paper has the highest classification accuracy. Through verification and comparison, the proposed algorithm can segment defects quickly and accurately and suppress noise interference effectively. It is better than other traditional image threshold segmentation methods, validated by both segmentation accuracy and computational efficiency. At the same time, the real-time performance of our algorithm was performed on the advanced SEED-DVS8168 platform.

## 1. Introduction

With the development of automobiles and aerospace, military and national defense, microelectronics industry, modern medicine, bioengineering and instrumentation, and other industries, the demand for small magnetic rings is increasing drastically [1]. However, the surface of the magnetic ring is prone to physical defects such as scars, cracks, and trachoma due to the influence of impurities mixed with raw materials, damage to the forming mold, and unhomogenzation heating during firing [2]. These defects affect the operating efficiency and service life of the parts themselves and their products directly [3]. It also reduces the stability and reliability of the equipment and brings great hidden dangers to using it safely. Therefore, it has become an urgent problem to be solved: How to improve the efficiency and accuracy of magnetic ring defect detection, in particular, those subtle defects invisible to the eye or by an ordinary CCD camera [4]. Image processing is the primary link in defect detection. Images with noise pollution must undergo preprocessing operations such as grayscale correction, filtering, and noise reduction before they are used for image processing [5]. In image processing technology, image segmentation is an important method to extract regions of interest [6]. Common image segmentation methods include edge detection and threshold segmentation. The information detected by the edge detection method is too strong, and the edge information is too weak to achieve the ideal segmentation effect. The threshold value determined by the threshold segmentation method, such as the selection iteration method and the maximum inter-class variance method (OTSU), is larger, and the segmentation effect of them is poor [7]. The most commonly used method is still the maximum entropy method. Although the segmentation effect of dark areas is slightly better, it still cannot segment the defect area and the background area accurately [8].

In order to solve the problem of accurate segmentation and calculation rate of fine defect images on the surface of small magnetic rings, this study introduces the 2D-Gabor filter using common scar defects as an example and proposes a new adaptive threshold segmentation method based on the improved 2D-Gabor filter bank. This method has the characteristics of low entropy, multi-resolution, and decorrelation, and can be used to filter image noise from different scale directions to remove normal textures, so as to achieve accurate segmentation of small magnetic ring surface defects. We designed a classifier based on a BP neural network to classify the scar images and crack images determined by the adaptive threshold segmentation method and the traditional threshold segmentation method, so as to verify the effectiveness of the method. At the same time, it realized the real-time nature of this algorithm on the DVS8168 platform. The flowchart of this study is shown in Figure 1.

## 2. Methods and Results Analysis

### 2.1. Raw Magnetic Ring Image Acquisition

The image quality obtained by the imaging system is crucial for post-image processing. In order to obtain a clear image of the surface defects of the magnetic rings, we obtained 10 qualified magnetic rings and 150 defective magnetic rings (Shandong Tongfang Luying Electronics Co., Ltd., Shandong, China). The height of the magnetic ring is about 30 mm, the diameter is about 20 mm, and there is a fine crack width of about 0.1 mm on the surface. The magnetic ring is small in size and has a fine and irregular texture, which is difficult to observe with the human eye and general optical imaging devices, especially the radial microcracks submerged in the normal texture. Therefore, a microscope was designed in this study, which is shown in Figure 2.

The camera used in the experiment had a microscope lens magnification of 10×, a focal length of 5 cm, and a resolution of 1 K. The light source is a stable and economical LED lamp (Opple Lighting Co., Ltd., Shanghai, China), the light intensity is 500~700 lux, the illumination mode is diffuse, and a 768 × 576 dot matrix can be collected. In order to avoid interference from natural light, turn off the room lights before the experiment, turn on the microscope, light source, and computer, and then place the magnetic ring on the rotating stage of the translation stage, which not only realizes millimeter-level translation in the horizontal direction but also realizes the full detail capture of the magnetic ring surface. By adjusting the position of the translation stage appropriately, the camera achieves the best depth of field and field of view.

When acquiring defect images, first, adjust the position of the translation table according to the defect markers so that the camera is roughly aimed at the defect area. Then, the magnetic ring surface defect is looked for by slowly rotating the rotary table. Then fine-tune the distance to the focal length to ensure that the microscope camera captures a clear, complete, and bright image of the defect. Finally, the image information is stored in the computer. If the defect area is large, slowly move the shooting area along the defect texture, starting from the edge point of the defect, so that one edge of the defect coincides with the opposite edge of the previous image. As can be seen in Figure 3, the image acquisition device is able to clearly detect defects submerged in the texture.

### 2.2. Image Processing

The accuracy requirement of image segmentation is very important to find the subtle defects. The Gabor filter was customly applied. The Gabor function was first proposed by Dennis Gabor in 1946 [9]. And in 1985, Daugman pushed the 1D-Gabor filter up to two-dimensions successfully [10]. The 2D-Gabor filter is a linear filter that can achieve local optimal solutions in both the spatial and frequency domains [11]. The representation of its frequency and direction is very close to the human visual system [12], so it is often used for texture description. In addition, Gabor filters have self-similarity, and they can be generated from a mother wavelet through dilation or rotation.

Different parameter values can be obtained from 2D-Gabor filters with different bandwidths, frequencies, and directions. We can use these to filter the target image and extract the image texture feature information with a certain direction and changing law. The 2D-Gabor function can be interpreted as the product of an elliptic Gaussian envelope and a complex plane wave [13], and is interpreted in the following Formula (1):(1)$$\Psi_{\left(s,d\right)}(x,y)=\Psi_{k}(v)=\frac{{\left||k|\right|}^{2}}{\sigma^{2}}\cdot exp(\frac{{\left||k|\right|}^{2}\cdot {\left||v|\right|}^{2}}{2\sigma^{2}})\cdot \left[exp(ik\cdot v)-exp(-\frac{\sigma^{2}}{2})\right]$$
where v=(x,y) is the position variable, σ is the window size space constant, $k=k_{s}\cdot exp(i_{\Phi_{d}})$ is the frequency vector, which represents the scale and orientation of the 2D-Gabor filter. $k_{s}=k_{max}/f^{s}$ indicates the center frequency of the Gabor filter.

In Formula (1), the spatial parameter σ determines the bandwidth of the filter, and the value is π; *k* defines the key parameters $k_{s}$ and $\Phi_{d}$ for the value of the Gaussian function in this study. It has been shown that the experiment works best when the center frequency does not exceed π/2 in the experiments conducted by M Lades. Therefore, this study takes the maximum sampling frequency $k_{max}=\pi /2$, and the sampling step size $f=\sqrt{2}$, s=0,1,2,3. Since the Gabor filter is symmetric, the actual value of $\Phi_{d}$ is between [0, π]. Therefore, the value of $\Phi_{d}$ is πd/8, d=0,1,...,7. Thus, a total of 32 real part value images of Gabor filters in 4 scales and 8 directions are formed in Figure 4.

### 2.3. Filtering of Scar Defect Images

The process method of using the Gabor filter bank to process the magnetic ring defect image can be interpreted in the Formula (2).
(2)$$g(x,y)=|f(x,y)*\psi_{s,d}(x,y)|$$
where f(x,y) represents the input image, and g(x,y) represents the output image after processing. $\Psi_{s,d}(x,y)$ is the filter template, and the size of the filter template is selected as (15, 15) in this study. The symbol “∗” stands for convolution operation. The original defect image is processed using the above filter bank, and the processed images are shown in Figure 5. Different scale parameters are represented from top to bottom (s=0,1,2,3), and different direction parameters are represented from left to right $\left(k_{max}=0,\pi /8,...,7\pi /8\right)$.

In Figure 5, the larger the filter scale, the stronger the suppression of small noise. In addition, the filtering result changes with the filter direction transformation, and the texture response value to a direction close to the filter direction is larger. The magnetic ring surface images involved in this study all have normal textures generated during production and processing in the horizontal direction. Therefore, the original filter bank is improved, which means that the horizontal direction filter $\left(\Phi_{d}=\pi /2\right)$ is screened out to avoid the adverse effect of normal texture on the segmentation of defective areas. The edge direction of the defect area in the magnetic ring image is obvious, and the horizontal texture of the background area has been filtered out. At the same time, other speckle noise appears isotropic with no well-directed edges. Therefore, the isotropic filter response can be suppressed by stacking the filtered images to enhance the difference between the defect and the background area. The superposition process is described in Formula (3).
(3)$$g(x,y)=|\sum \sum f(x,y)*\Psi_{s,d}(x,y)|$$
where f(x,y) and $\Psi_{s,d}(x,y)$ represent the same meanings as that in Formula (2), and g(x,y) represents the superimposing of the output images after processing. Different images, including both initial images and processed images, are indicated in Figure 6.

In Figure 7, it can be clearly seen that there are fewer and fewer red areas. This shows that the improved filter set makes the grayscale more uniform, effectively weakens the noise pollution in the image, and makes the difference between defects and background more prominent. Therefore, the improved filter bank processing works better.

In practical terms, in order to simplify the complex convolution operation process, filters with the same window and different scales and directions are usually superimposed first, and then the image is filtered. The process can be expressed in Formula (4):(4)$$g(x,y)=|f(x,y)*\sum_{s}\sum_{d}\Psi_{s,d(x,y)}|$$
where g(x,y), f(x,y)*,* and $\Psi_{s,d}(x,y)$ represent the same meanings as that in Formula (3).

Here, the image size is assumed to be M×N, and the filter template size is H×W. Thus, the calculation amount is S(28M×N×H×W) by Formula (3). The calculation amount of the simplified Formula (4) method is: S(28H×W+M×N×H×W)≈S(H×W×M×N) by the simplified Formula (4). Obviously, the calculation was speed by more than one order of magnitude.

The difference between the target defect and the background region is strengthened, which provides preparation for the implementation of adaptive threshold segmentation. The grayscale histogram of the image can clearly reflect the grayscale statistical law after passing through the filter bank.

In order to verify the filtering effect of the improved filter bank on other defective images, we used the improved filter bank to filter the cracks. The filtering effect is also obvious, as shown in Figure 8.

By the grayscale statistical characteristics, it is indicated that the grayscale histogram of the filtered defect image is dominated by the background pixels that approximately obey the normal distribution. The defect area is represented by a small number of low-gray pixels, and there is an obvious boundary with the pixels in the background area. According to this characteristic, the segmentation threshold can be constructed using Formula (5).
(5)T=μ−k×σ
where μ and σ represent the gray mean and variance of the filtered enhanced image, respectively, and *k* is the adjustment factor. The size of *k* is related to the gray value characteristic of the enhanced image, and this characteristic can be described by the gray mean value. Through extensive experiments, we define *k* as a function of the image mean in a linear Formula (6).
(6)k=1.0+[1−exp(0.60−0.0045μ)]/ [1+exp(0.60−0.0045μ)]

### 2.4. Adaptive Threshold Segmentation Results

For filter-enhanced image pixels  g(x,y), the rules for defining adaptive threshold segmentation are illustrated in Formula (5).
(7)$$g(x,y)=\left\{\begin{array}{c} 0(Background\;region),\;\;\;g(x,y)>T\\ 1\;\;\;\;\;\;(Defect\;region),\;\;\;g(x,y)<T \end{array}\right.$$

Traditional methods of threshold segmentation, such as the common selection iterative method, OTSU method, and maximum entropy method et al., are selected to segment the threshold of a magnetic ring image after Gaussian filter noise reduction, and the results are separately shown in Figure 9a–c.

On the left of Figure 10, the processing result is displayed by the threshold segmentation algorithm proposed by this study. Although there is a slight mis-segmentation, it can be eliminated by the area picking method, as shown in the middle of Figure 10. Finally, after adaptive segmentation and morphological processing based on a 2D-Gabor filter, the magnetic ring scar defect is basically detected. By detecting the minimum circumscribed rectangle of the defect, the defect area of interest can be enveloped, as shown in the right of Figure 10.

The segmentation thresholds, processing times, and the number of iterations determined by different threshold segmentation algorithms for scar defect images are labeled in Table 1.

In Table 1, it can be indicated that our algorithm can accurately separate the defect area from the background area without iterative time. However, the threshold determined by the iterative method and the OTSU method is relatively large. And the segmentation effect of the scar defect on the surface of the magnetic ring is poor. The threshold determined by the maximum entropy method is slightly smaller, as is a certain segmentation effect on dark areas with small defect gray values, especially for large-area defects such as scars. But it cannot be completely segmented.

In order to further verify the superiority of the adaptive threshold segmentation algorithm proposed in this paper compared with other traditional threshold segmentation algorithms, a classifier based on a BP neural network was designed. By comparing the classification results of classifiers under different threshold segmentation methods, the superiority of the proposed algorithm is verified.

## 3. Classification of Magnetic Ring Surface Scars and Cracks Based on BP Neural Network

### 3.1. BP Neural Network Structure and Parameter Design

Before establishing the BP neural network classification model, the number of neurons in the input, output, and hidden layer, learning rate, number of iterations, allowable error value, and inertia coefficient should be set in advance.

The number of input nodes can be the number of pixels of the image or the number of feature dimensions. The number of output nodes is generally based on the number of categories to be divided, and if you want to divide the data into m categories, the number of nodes is generally m or ${log}_{2}^{m}$. In this example, the number of input nodes is set to the number of image pixels, and the number of output nodes is set to 2 according to the scar and the crack that we want to recognize. The number of nodes in the hidden layer directly affects the classification accuracy of the model. If the number of nodes is too small, the network training may not be completed, which will affect the learning efficiency. Too much can prolong the training time or even lead to a failure to converge. In practice, the number of hidden layer nodes is determined using Formula (8).
(8)$$l={log}_{2}^{m}(\sqrt{m+n}+a<l<n-1)$$
where n, m, and l are, respectively, the number of input, output, and hidden layer nodes, and a is a constant less than 10.

The learning rate η affects the correction amount of each weight, and too small will lead to slower convergence speed and increase the transmission error. Too much will affect the system’s performance. According to experience, the value of η is generally between [0.01, 0.8]. The node excitation function also affects the performance of the BP neural network classifier. Common node excitation functions include threshold, S-type, and Gaussian stimulus functions. Among them, the network structure of the S-type function is more applicable and can be approximated to any function, so the stimulus function in this paper is the S-type function.

### 3.2. Analysis of Classification Results

Table 2 shows the experimental environment settings. The settings of the relevant parameters of the BP neural network model are shown in Table 3.

The number of nodes in the hidden layer can be obtained in the range of (4, 14) according to Formula (8), and the accuracy of defect classification under different hidden layer nodes is obtained through repeated experiments, as shown in Figure 11.

According to the results of Figure 11, it is concluded that when the number of neurons in the hidden layer of the BP neural network is 10, the overall classification accuracy of the sample is the highest. Finally, Table 4 shows the specific classification results of the simulation test of scars and cracks samples using the model designed with the above parameters.

As can be seen from Table 4, the average accuracy of the adaptive threshold segmentation method proposed in this paper is as high as 97.5%, which is 12.5% higher than that of the iterative method, 10% higher than the OTSU method, and 2.5% higher than the maximum entropy method. It is proven that the adaptive threshold segmentation method proposed in this paper is superior to other traditional threshold segmentation methods in terms of classification ability.

## 4. Real-Time Implementation

### 4.1. The Overview of DVS8168

The inspection of magnetic ring image defects is based on DVS8168, which was developed with the TMS320DM8168 processor at its core. This device has a strong ability to process video; the embedded ARM core can be clocked at a frequency of 1.2 GHz, and the embedded DSP core can reach 1 GHz, mainly for video compression and high-speed data processing. The high-definition video processing subsystem (HDVPSS) includes three high-definition video coprocessors (HDVICP) at frequencies up to 600 MHz. The high-definition video processing subsystem, HDVPSS, is controlled by Cortex-M3, which mainly completes the video capture task. SEED-DVS8168 peripherals include: 8 chips of 128 MB DDR3 with a total of 1GB of memory, 16-bit bus width, 256 MB of NAND Flash memory, and HDMI as a high-definition video output. The input port enables 16 D1 images and comes with a 160 GB hard disk for storing video processing data. The A8 core is the main processor, responsible for the control and external interface of the whole system. The C674xDSP is responsible for algorithm calculation, the Video M3 is responsible for the encoding and decoding of video, the VPSS M3 is responsible for the video processing subsystem, and the SATA bus stores and accesses the compressed video and communicates with the host computer through the driver setting serial port. The overall structure of the system is shown in Figure 12.

### 4.2. The Principle of Magnetic Ring Detection System

The magnetic ring detection system is mainly composed of three parts: Industrial camera and microscope, SEED-DVS8168, and PC. The algorithm for magnetic ring defect detection is embedded in SEED-DVS8168, and the real-time display interface of the system is shown in Figure 13.

After the system is powered on, the HDVPSS, controlled by Cortex-M3, acquires a magnetic ring image from CCD industrial cameras and microscopes. HDVICP is a video codec accelerator that can independently complete video encoding and decoding and has obvious advantages in processing high-definition video. Then, the processing of image data is completed in the DVS8168 main core; its frequency is up to 1 GHz, and its processing performance is as high as 8000 MI/s, which greatly improves the speed of data processing. Finally, under the control of the ARM core, the data are read, stored, and transmitted to the PC through HDMI for real-time display. This magnetic ring detection system can meet the real-time display of magnetic ring images with 450 × 450 resolution and a 50 Hz frame rate, and the system can achieve a speed-up ratio that is nearly 4 times faster than running in MATLAB. This system processes an image for about 18 ms, which is a good improvement in the visual effect of the real-time display of magnetic rings.

## 5. Conclusions

The image processing method of surface defects of small magnetic rings is mainly studied in this study. We designed a microscope instead of the traditional CCD camera to collect the initial image, which greatly improved the acquisition efficiency of subtle defects. In the case of the poor performance of common threshold segmentation algorithms, a new adaptive threshold segmentation method was proposed based on an improved 2D-Gabor filter bank. In addition, a classifier based on a BP neural network was designed to classify the scar images and crack images determined by the adaptive threshold segmentation method and the traditional threshold segmentation method. By comparison, it has faster calculation speed and higher classification accuracy than traditional methods, such as selection iteration, OTSU, and maximum entropy. And an advanced DVS8168 video platform is used to implement the above-mentioned algorithm in real time and verify its feasibility. This study might be beneficial to automatic sorting for sublet defects on the surface of small magnetic rings.

## Figures and Tables

**Figure 1 sensors-24-01031-f001:**
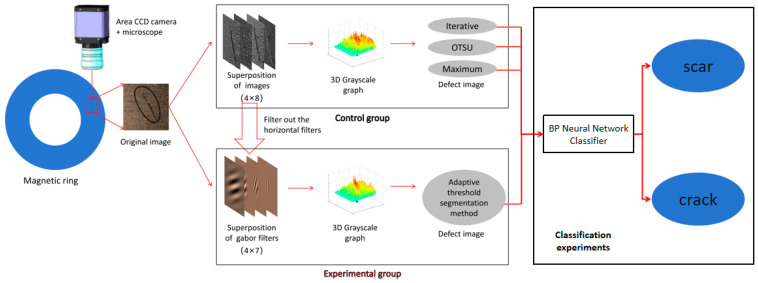
Flow diagram of image segmentation in this study.

**Figure 2 sensors-24-01031-f002:**
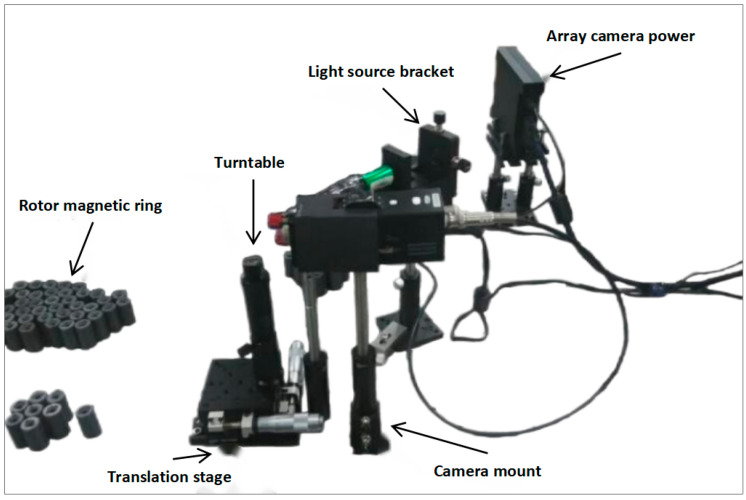
Original magnetic ring image acquisition system.

**Figure 3 sensors-24-01031-f003:**
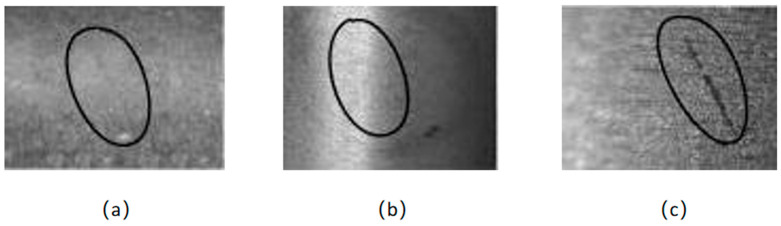
Physical imaging comparison: (**a**) the ordinary camera; (**b**) the CCD camera; (**c**) area CCD camera + microscope.

**Figure 4 sensors-24-01031-f004:**
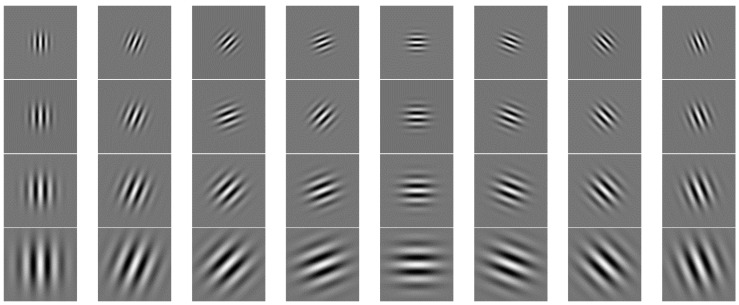
Effect diagram of real part value of 2D-Gabor filter with different scales and directions.

**Figure 5 sensors-24-01031-f005:**
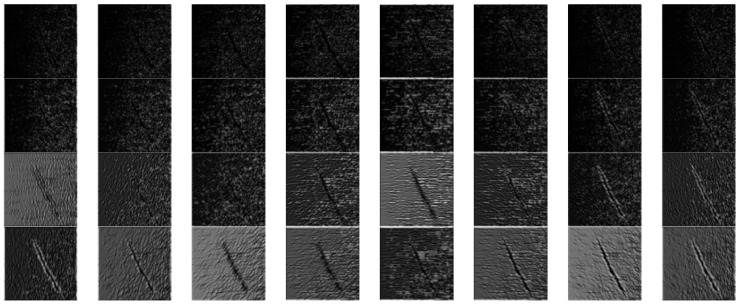
Two-dimensional (2D)-Gabor filtering images with subtle defect at different scales and directions.

**Figure 6 sensors-24-01031-f006:**

The initial image and processed images by 2D-Gabor filters: (**a**) initial image; (**b**) filter image accumulation of 4 × 8 filter banks; (**c**) filter image accumulation of improved filter banks.

**Figure 7 sensors-24-01031-f007:**
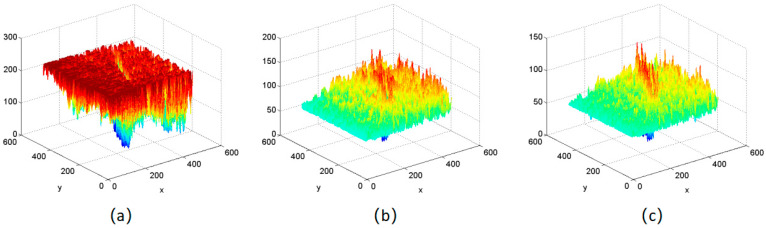
The corresponding 3D grayscale images. (**a**) initial image; (**b**) filter image accumulation of 4 × 8 filter banks; (**c**) filter image accumulation of improved filter banks.

**Figure 8 sensors-24-01031-f008:**
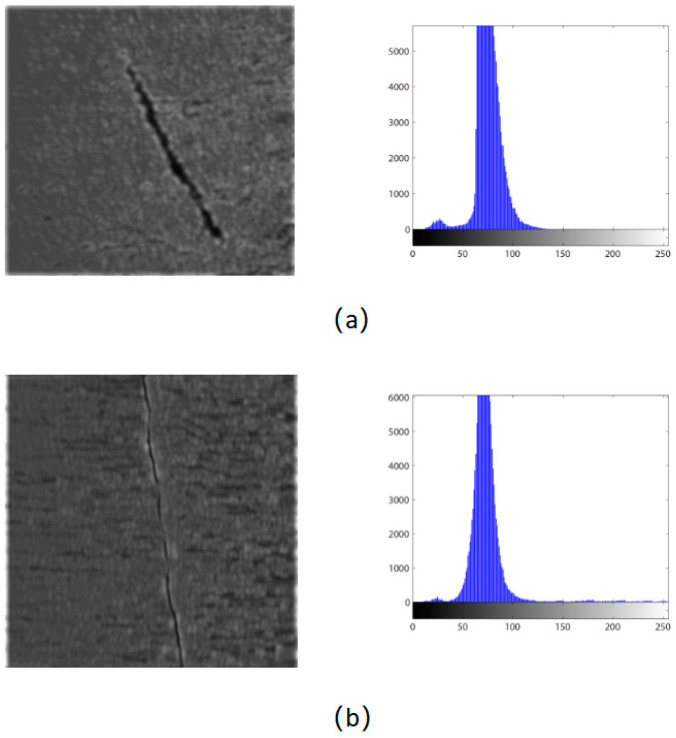
Improved 2D-Gabor filter bank processing image and gray histogram: (**a**) scar; (**b**) crack.

**Figure 9 sensors-24-01031-f009:**
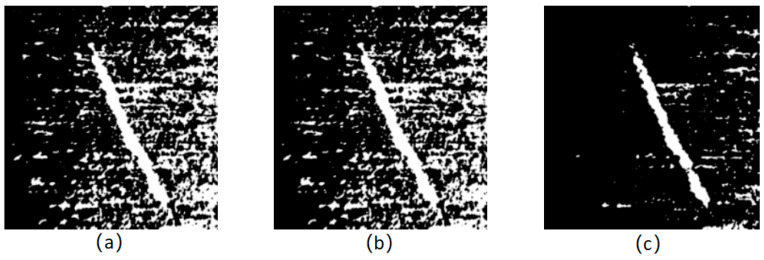
Segmentation results of traditional methods. (**a**) Iterative; (**b**) OTSU; (**c**) maximum entropy.

**Figure 10 sensors-24-01031-f010:**
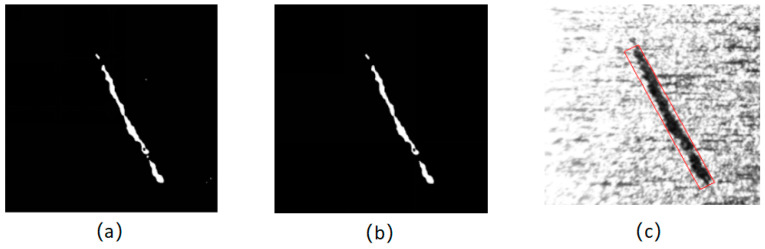
Thresholding segmentation results of scar defect images. (**a**) Adaptive threshold segmentation result; (**b**) area pick-up result; (**c**) defect image.

**Figure 11 sensors-24-01031-f011:**
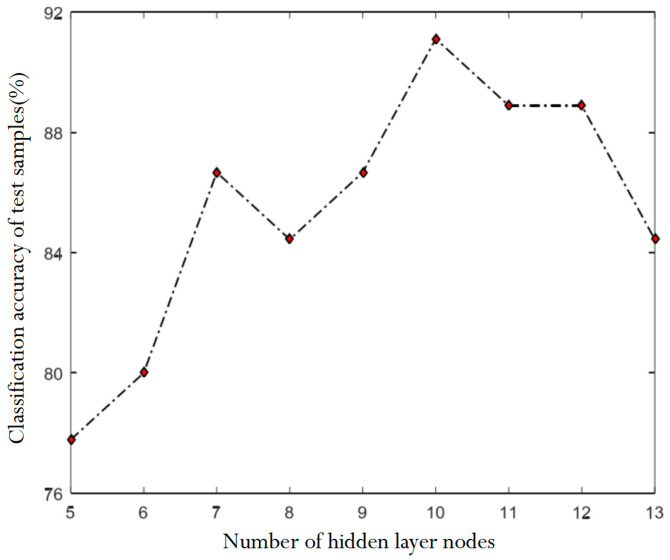
Classification accuracy under the number of neurons in different hidden layers.

**Figure 12 sensors-24-01031-f012:**
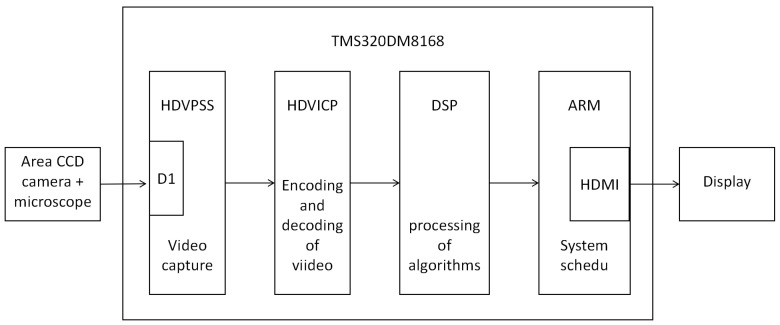
Structural diagram of magnetic ring detection system.

**Figure 13 sensors-24-01031-f013:**
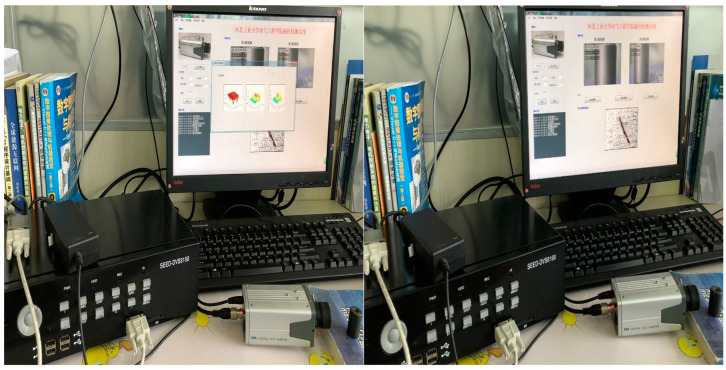
Real-time display of processing results.

**Table 1 sensors-24-01031-t001:** Comparison of 4 kinds of threshold segmentation algorithms.

Method	Defect Images		Number of Iterations
	Threshold	Processing Time	
Iterative	166	336	256
OTSU	157	297	6
Maximum entropy	127	60	8
Adaptive threshold	95	51	0

**Table 2 sensors-24-01031-t002:** Experimental operating environment.

Project	Disposition
Processor model	Intel(R) Core(TM) i5-4210M
Operating system	Windows 10
CPU frequency	2.60 GHz
RAM	8.00 GB
Experimental platform	Matlab R2020a

**Table 3 sensors-24-01031-t003:** Model parameter value.

Model Parameters	Value
The number of input nodes	10
The number of nodes in the hidden layer	L
The number of output nodes	3
The number of iterations	20,000
Error accuracy	0.5%
The learning rate	15%

**Table 4 sensors-24-01031-t004:** Classification results of scars and cracks determined by different threshold segmentation methods.

Threshold Segmentation Method	Type of Defect	Number of Training Samples	Number of Test Samples	Number of Identification Errors	Classification Accuracy	Average Accuracy
Iterative	Scar	50	20	2	90%	85%
	Crack	50	20	4	80%	
OTSU	Scar	50	20	1	95%	87.5%
	Crack	50	20	4	80%	
Maximum entropy	Scar	50	20	0	100%	95%
	Crack	50	20	2	90%	
Adaptive threshold	Scar	50	20	0	100%	97.5%
	Crack	50	20	1	95%	

## Data Availability

The data presented in this study are available on request from the corresponding author.

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
