# Peer review of "Adaptive Segmentation Algorithm for Subtle Defect Images on the Surface of Magnetic Ring Using 2D-Gabor Filter Bank"

_sensors, 2024, doi:10.3390/s24031031_

Round 1

Reviewer 1 Report

Comments and Suggestions for Authors

Following are my comments and suggestions:

1. Title needs to be changed as:
"Adaptive Segmentation Algorithm for Subtle Defect Images on The Surface of Magnetic Ring using ....."

Please mention the name of the method in the above suggested title.

2. Abstract: Rewrite the following sentence:

The segmentation threshold of the algorithm proposed in this paper is 40% lower than that of iterative and 18 OTSU method and 25% lower than that of maximum entropy method, and the processing time is shorter.

3. Include flowchart of the method.

4. Source of dataset is not included in the study. Provide the source.

5. Table 1: More comparative study is required.

6. There are many segmentation algorithms. The authors can review them: https://doi.org/10.1016/j.knosys.2021.107432

Reviewer 2 Report

Comments and Suggestions for Authors

The paper proposes a new adaptive threshold segmentation method for subtle defect images on the surface of small magnetic rings, based on the improved 2D-Gabor filter bank. 

The paper designs an acquisition system for a microscope to capture the initial image of the magnetic ring defects, which improves the acquisition efficiency of subtle defects that are invisible by ordinary CCD camera or eyes. The paper compares the proposed method with three traditional threshold segmentation methods (iterative, OTSU, and maximum entropy), and shows that the proposed method has faster calculation speed and more obvious segmentation effect. The paper implements the proposed method in real time on the advanced DVS8168 video platform, and verifies its feasibility and practical significance for the automatic sorting of subtle defects on the surface of small magnetic rings.

I am somewhat uncertain regarding the authors' intention in the phrase 'a microscope was designed' (page 2, line 53). It appears to me that the authors intended to convey the design of the acquisition system. If this is the case, it must be rephrased to make it clear. Furthermore, the explication concerning the designed component, be it the microscope or the acquisition system, is exceedingly limited, rendering it insufficient for the reproduction of the results.

The paper fails to elucidate in detail the manner in which the 2D-Gabor filter bank has been enhanced, the parameters employed, and the methodology for their determination. The paper also does not provide any visual examples of the filtered images or the extracted features. This makes it hard to understand the working mechanism and the advantages of the improved 2D-Gabor filter bank. 

The paper limits its examination to a singular type of defect (a fine crack) on a solitary type of object (a small magnetic ring). The paper does not discuss how the method can be applied to other types of defects, objects, or scenarios. The paper also does not consider the possible variations or noises in the image acquisition process, such as lighting, angle, resolution, or distortion. This limits the practical applicability and reliability of the method.

I understand the focus of the paper has been on the thresholding methods. Nevertheless, I maintain that it is imperative to provide experimental results or analyses for the purpose of comparing the proposed method with other existing methods of image segmentation, such as deep learning or region-based methods. This makes it difficult to evaluate the effectiveness and novelty of the proposed method. Should such comparisons not be included, it is incumbent upon the authors to explicitly state this, bolstered by robust arguments to justify their choice.

Round 2

Reviewer 2 Report

Comments and Suggestions for Authors

Thanks for updating the manuscript, and I am now inclined to recommend its acceptance.